# Contamination of Fresh Produce with Antibiotic-Resistant Bacteria and Associated Risks to Human Health: A Scoping Review

**DOI:** 10.3390/ijerph19010360

**Published:** 2021-12-30

**Authors:** Mahbubur Rahman, Mahbub-Ul Alam, Sharmin Khan Luies, Abul Kamal, Sharika Ferdous, Audrie Lin, Fazle Sharior, Rizwana Khan, Ziaur Rahman, Sarker Masud Parvez, Nuhu Amin, Rezaul Hasan, Birkneh Tilahun Tadesse, Neelam Taneja, Mohammad Aminul Islam, Ayse Ercumen

**Affiliations:** 1Environmental Interventions Unit, Infectious Disease Division, International Centre for Diarrhoeal Disease Research, Dhaka 1212, Bangladesh; mahbubalam@icddrb.org (M.-U.A.); sharminkhan@icddrb.org (S.K.L.); abul.kamal@icddrb.org (A.K.); sharika.ferdous@icddrb.org (S.F.); fazle.sharior@icddrb.org (F.S.); rizwana.khan@icddrb.org (R.K.); ziaur.rahman@icddrb.org (Z.R.); parvez@icddrb.org (S.M.P.); nuhu.amin@icddrb.org (N.A.); rezaul.hasan@icddrb.org (R.H.); 2Berkeley’s School of Public Health, University of California Berkeley, Berkeley, CA 94720, USA; audrielin@berkeley.edu; 3Faculty of Medicine, The University of Queensland, Brisbane, QLD 4072, Australia; 4Institute for Sustainable Futures, University of Technology Sydney, 235 Jones St., Ultimo, NSW 2007, Australia; 5School of Medicine, Hawassa University, Shashemene, Awassa P.O. Box 5, Ethiopia; birknehtilahun@gmail.com; 6International Vaccine Institute, SNU Research Park, 1 Gwanak-ro, Gwanak-gu, Seoul 08826, Korea; 7Department of Medical Microbiology, Postgraduate Institute of Medical Education and Research, Chandigarh 160012, India; drneelampgi@yahoo.com; 8Allen Center, Paul G. Allen School for Global Health, Washington State University, 240 SE Ott Road, Pullman, WA 99164, USA; amin.islam@wsu.edu; 9Department of Forestry and Environmental Resources, North Carolina State University, 2800 Faucette Drive, 3120 Jordan Hall, Raleigh, NC 27607, USA; aercume@ncsu.edu

**Keywords:** antimicrobial resistance, antibiotic-resistant bacteria, antibiotic resistance genes, agriculture, fresh agriculture products, vegetables, fruits, leafy greens, retail markets, health risks

## Abstract

Fresh produce, when consumed raw, can be a source of exposure to antimicrobial residues, antimicrobial-resistant bacteria (ARB) and antimicrobial resistance genes (ARGs) of clinical importance. This review aims to determine: (1) the presence and abundance of antimicrobial residues, ARB and ARGs in fresh agricultural products sold in retail markets and consumed raw; (2) associated health risks in humans; and (3) pathways through which fresh produce becomes contaminated with ARB/ARGs. We searched the Ovid Medline, Web of Science and Hinari databases as well as grey literature, and identified 40 articles for inclusion. All studies investigated the occurrence of multidrug-resistant bacteria, and ten studies focused on ARGs in fresh produce, while none investigated antimicrobial residues. The most commonly observed ARB were *E. coli* (42.5%) followed by *Klebsiella* spp. (22.5%), and *Salmonella* spp. (20%), mainly detected on lettuce. Twenty-five articles mentioned health risks from consuming fresh produce but none quantified the risk. About half of the articles stated produce contamination occurred during pre- and post-harvest processes. Our review indicates that good agricultural and manufacturing practices, behavioural change communication and awareness-raising programs are required for all stakeholders along the food production and consumption supply chain to prevent ARB/ARG exposure through produce.

## 1. Introduction

The World Health Organization (WHO) and Food and Agriculture Organization (FAO) promote the daily consumption of fruits and vegetables as part of a healthy diet, due to their high nutritional value [1,2]. Raw consumption of many fresh leafy and non-leafy vegetables, root vegetables, sprouts, and fruits results in the exposure of humans to foodborne bacterial pathogens, including antibiotic-resistant bacteria (ARB) [3,4,5,6]. In recent decades, exposure to antimicrobial-resistant pathogens through the food chain has increasingly been reported to cause foodborne disease outbreaks [7].

Studies have reported the presence of ARB and antibiotic resistance genes (ARGs) on fresh produce. For example, in Japan, extended-spectrum β-lactamase (ESBL)-producing pathogens have been found on fresh produce [8]. Multidrug-resistant *E. coli* and *Salmonella* spp. on vegetables have been linked with disease outbreaks in Germany, the United States, Canada, Australia and Finland [9,10,11,12,13]. Additionally, opportunistic microorganisms previously considered non-pathogenic are present in fresh produce, and can cause serious infections in an immune-compromised host. For example, opportunistic bacteria, such as *Klebsiella* spp. and *Enterobacter* spp., have been found on vegetables (e.g., cabbage, capsicum and tomatoes) in retail markets in different settings [14,15,16]. Consumption of vegetables contaminated with *Klebsiella* spp. can cause acute bronchopneumonia and labor pneumonia in immunocompromised individuals [17].

Fresh produce can be contaminated with bacterial pathogens at multiple points throughout its production and supply chain by direct contact with fecal waste during farming, such as wastewater irrigation and the use of biosolids or animal manure as fertilizer [18,19]. Contamination can also happen during the transport and handling of produce [20]. While these potential contamination pathways have been studied well for traditional pathogens, their relative contributions to the contamination of fresh produce with ARB, ARGs and antimicrobial residues have not been quantified. Such information could potentially inform interventions to reduce human exposure to ARB/ARGs through fresh produce. Therefore, this scoping review aimed to synthesize data on the presence and abundance of ARB, ARGs and antimicrobial residues present on fresh produce that are sold in retail markets and typically consumed raw. In addition, the health effects in humans due to the consumption of fresh produce contaminated with ARB/ARGs, and the pathways through which fresh produce becomes contaminated with ARB/ARGs, were investigated.

## 2. Methods

We developed a protocol that specified the research questions, inclusion/exclusion criteria, data sources and search engines, and followed the Preferred Reporting Items for Systematic reviews and Meta-Analyses extension for Scoping Reviews (PRISMA-ScR) checklist to conduct the review [21] (Appendix A).

### 2.1. Data Sources

Two reviewers independently searched the Ovid Medline, Web of Science and Hinari databases for peer-reviewed literature published from 1 January 2001 to 18 October 2020. We searched Google, Google Scholar and ProQuest for grey literature, and manually searched the reference list of the included articles for additional relevant publications.

### 2.2. Search Strategies

We conducted a preliminary search for published scientific literature on the topic of interest to identify keywords for conducting an advanced search. We developed key search terms for three domains, including: (a) antimicrobial resistance; (b) agriculture and fresh agricultural products; and (c) site of sample collection (Table 1). Two reviewers primarily formulated the search strategy, with feedback from a third reviewer. The search strategy used for Medline is attached in the Appendix A.

### 2.3. Screening, Data Extraction and Synthesis

According to the PRISMA guidelines, we selected articles in four phases: (i) identification; (ii) screening; (iii) eligibility; and (iv) inclusion. The literature identified by the search terms was imported to EndNote (version X9) and duplicates were removed. The updated list was then imported to Rayaan online software. Four researchers collaboratively reviewed the articles by screening the titles and abstracts according to our inclusion and exclusion criteria (Table 1), and shortlisted articles were screened in full text. Disagreements regarding eligibility were resolved through discussions among the reviewers, with approval from a third reviewer.

We then extracted and compiled the results from the eligible studies using a data extraction matrix to organize data. This data matrix was disaggregated into four key themes with relevant sub-themes, including (i) presence and/or abundance of ARBs, ARGs and antimicrobial residues on fresh produce (vegetables and fruits consumed in raw condition) sold in the retail markets; (ii) public health risks from consuming raw fresh produce in relation to AMR; (iii) pathways of contamination of fresh produce with ARB/ARGs/antimicrobial residues; and (iv) actions recommended to reduce contamination of fresh produce with ARB, ARGs and antimicrobial residues (Table 2). Two reviewers independently extracted data on each theme. We also extracted data on study location, methods, year and duration, the number of samples investigated, number of isolates found, the method used for antibiotic susceptibility testing and the name of specific serotypes and genes that confer resistance to antibiotics, where available.

Due to heterogeneity of the study design, sample types, and different breakpoints used to determine antibiotic susceptibility, we performed a narrative analysis to synthesize the findings from the included studies. The key results of the narrative analysis were used to group the data into themes.

## 3. Results

After duplication removal, the literature search from various databases, including the grey literature, yielded 146 unique articles, of which 40 articles met the inclusion criteria of our scoping review (Figure 1). Among these, 37 were laboratory-based peer-reviewed research articles, including three short communication articles. The grey literature included one report and two research theses. Overall, 37 of the included articles were cross-sectional studies, and three were longitudinal studies. The studies were conducted in diverse locations, including both high- and low-income counties in Asia (Bangladesh, India, Pakistan, China, Korea, Malaysia), the Americas (Mexico, USA, Canada), Europe (Czech Republic, Portugal, Germany, Italy) and Africa (South Africa). All articles quantified ARB in raw vegetables and fruits, while only 10 detected ARGs. We did not find any articles that detected or quantified antimicrobial residues on vegetables found in retail markets. Twenty-five articles discussed the human health risks of consuming raw fresh produce carrying ARB. Moreover, 22 articles included discussions of the pathways of contamination of fresh produce with ARB. We excluded 106 articles, among which 62 were related to animal-based food, and 30 were not associated with AMR.

### 3.1. Prevalence of Antimicrobial-Resistant Bacteria on Fresh Produce

The included studies detected bacteria of more than twenty different genera on vegetables (leafy, non-leafy, root), fruits, sprouts consumed raw, and ready to eat (RTE) salad (Appendix A). The most frequently observed bacteria were *E. coli*, found in 17 out of 40 (42.5%) included articles. *E. coli*, including pathogenic strains, was mainly detected on lettuce, cabbage, cucumbers, and tomatoes. For example, among 260 samples of leafy, non-leafy and root vegetables investigated in Pakistan, approximately one-third of *E. coli* strains isolated from lettuce samples were identified as diarrheagenic *E. coli* pathotypes (DEPs) [22] (Appendix A). Pathogenic *E. coli* strains, such as Shiga toxin-producing *E. coli* (STEC), enteropathogenic *E. coli* (EPEC), and enterotoxigenic *E. coli* (ETEC) were detected in two studies, from nopalitos samples in Mexico [23] and from cucumber, lettuce and spinach samples in Pakistan [22]. DEPs in both studies reported resistance to multiple antibiotics including tetracycline (TET), ampicillin (AMP), ciprofloxacin (CIP), and gentamicin (GEN) (Appendix A). The study in Pakistan tested 50 DEPs isolates and found that 92% of the isolates were resistant to TET, 87% to AMP and 82% to efotaxime (CTX) [22]. Across the studies in our review, *E. coli* (not DEPs) isolated from fresh produce showed varying resistance to amoxicillin (AMX), AMP, GEN, erythromycin (ERY), colistin (CST), amikacin (AMK), cefotaxime (CTX) and ceftazidime (CAZ) (Appendix A). The resistance profiles for *E. coli* were diverse between the studies.

The second most common bacteria were *Klebsiella* spp. And *Salmonella* spp. *Klebsiella* spp., which were detected on produce in nine out of 40 articles (22.5%). Among these, *Klebsiella pneumoniae* was the most common; six out of nine articles (66.6%) detected this species from produce samples. For example, a study in China recovered 175 *Klebsiella pneumoniae* isolates from 216 samples of leafy, non-leafy, root vegetables and sprouts [24] (Appendix A). Another study in Algeria found 13 third-generation cephalosporin-resistant *Klebsiella pneumoniae* strains from 310 samples of fruits, leafy and non-leafy vegetables. Leafy vegetable lettuce was frequently contaminated. *Klebsiella* spp. was often resistant to aztreonam (ATM) and CTX [25]. CTX resistant *Klebsiella* spp. were also detected on onion, cucumber, tomato, chili pepper and ginger in India [26]. In Italy, *K. ozaenae* resistant to AMP, cefoxitin (FOX), CTX were detected in ready to eat (RTE) salad samples [27]. 

*Salmonella* spp. were found in nine out of 40 articles (22.5%). Lettuce samples were observed to be highly contaminated with *Salmonella* spp. followed by coriander, spinach, parsley, and sprouts. More than 100,000 samples of various types of fresh produce were investigated in the United States from 2002 to 2012, resulting in the detection of *Salmonella* spp. in 51 different serotypes of 152 samples [28]. Of the 51 *Salmonella* serotypes, 10 were resistant to different antibiotics [28]. The identified *Salmonella* serotypes were Oranienburg, Montevideo, Agona, Havana, Thompson, Poona, Kentucky, Tucson, Veneziana and one was unknown, with Tucson being the more prevalent. Antibiotic-resistant *Salmonella* serotypes Thompson, Poona, and Kentucky, were found on lettuce samples (23.1%), whereas the Oranienburg serotype was found on cantaloupe sample (6.7%) [28]. Another study in Malaysia isolated multidrug-resistant *S. enteritidis* from carrots that were resistant to AMP, AMX, trimethoprim (TMP) nalidixic acid (NA), trimethoprim-sulfamethoxazole (SXT), and chloramphenicol (CHL) [29]. In a study in Thailand, *S*. Stanley, *S*. Schwarzengrund and *S*. Rissen were isolated from lettuce samples, among which *S*. Schwarzengrund was resistant to AMP, CHL, NA, while *S.* Rissen was resistant to AMP, SXT and TET, and *S*. Stanley was resistant to NA [30]. Two studies in Malaysia found other *Salmonella* serotypes, such as *S*. Corvallis, *S*. Typhimurium and *S*. Enteritidis, detected from leafy vegetables, water dropwort, and long bean samples, which were accordingly resistant to multiple antibiotics [9,31] (Appendix A). A study in Mexico observed multidrug-resistant *Salmonella* in lettuce and carrot samples with resistance to AMP, cephalotin (CHT), CHL, TET, CIP, NA, streptomycin (STR), and SXT [32]. Resistance profiles of *Salmonella* isolates varied between studies and could not be compared, due to the detection of different serotypes. However, most of the studies (87.5%) found *Salmonella* isolates to be resistant to AMP (Appendix A).

Eight out of 40 articles (20%) reported the detection of *Pseudomonas* spp. in produce samples, and six of these studies reported isolates resistant to AMP. *Pseudomonas* spp. were frequently recovered from lettuce, carrots, and spinach. A study in Jamaica found that, among 88 *P. aeruginosa* isolates recovered from 95 vegetable samples, all were resistant to imipenem (IMP) (100%), followed by 97% to GEN, 93% to CIP, and 79% to CAZ [33].

Other ARBs that were often detected on leafy and non-leafy vegetables, fruits, and sprouts, were *B. cereus*, *Enterobacter* spp. [34] and *Listeria* spp. (predominantly *L. monocytogenes), Rahnella aquatilis* [35], *Staphylococcus* spp., *Shigella* spp. and *Citrobacter* spp. In addition, *Acinetobacter* spp. (predominantly *A*. *baumannii)*, *Sphingobacterium multivorum, Pseudomonas putida, Erwinia persicina, Pantoea agglomerans, Serratia fonticola,* and *Enterococcus* spp. *(E. faecalis* and *E. casseliflavus*) were also found on fresh produce (Appendix A). *Rahnella aquatilis* was infrequently isolated from spinach, whereas *Listeria* spp. was isolated from carrot and cabbage samples (Appendix A). *Enterococcus* spp. were also detected on produce. A study that tested 112 fruit samples in the USA found 16% of samples contaminated with enterococci [36]. Another study conducted in Germany found that all *B. cereus* strains isolated from 137 fresh vegetables were resistant against PEN G and CTX, and 99.3% were resistant to amoxicillin-clavulanic acid (AMC) [37]. For *Enterobacter* spp., one study in South Africa found that *E. cloacae* isolated from spinach, tomato and cucumber were resistant to aminoglycoside, CHL and TET, whereas another study from Italy reported *E. cloacae* strains isolated from frisée salad and RTE salad were resistant to AMP, AMC and CTX [27,38]. Two studies reported penicillin resistant *L. monocytogenes* in fresh produce, one in Malaysia which showed all 58 *L. monocytogenes* strains isolated from 301 vegetable samples were resistant to PEN G and 71% (*n* = 41) of isolates were resistant to meropenem [29]. Another study conducted in Brazil found that *L. monocytogenes* from raw and RTE salad vegetable samples were resistant to PEN G and TET [39]. RTE salads consisting of leafy and non-leafy vegetables without salad dressings were contaminated with antibiotic-resistant *E. coli* [22,27,40], *K. pneumoniae* [40], *L. monocytogenes* [39], *E. faecalis* and *E. faecium* [41] and *E. cloacae* [27].

### 3.2. Antibiotic Resistance Genes on Fresh Produce

Ten articles presented findings on ARGs. Of these, seven studies detected ARGs in *E. coli* isolated from fresh produce. For instance, a study conducted in the Czech Republic detected the ampicillin resistance gene *bla*_TEM_, and tetracycline resistance genes *tetA* and *tetB* in *E. coli* isolated from asparagus, rucola, leek and raddish samples [42]. In a study from China, plasmid-mediated mobile colistin resistance (*mcr-1*) gene was detected in *E. coli* isolated from an apple sample, which also carried ten more resistance genes including *aadA2, aadA1, floR, cmlA1, sul2, sul3, tetA, tetM, dfrA12, mdfA* [43] (Appendix A). *Mcr-1* positive *E. coli* were also isolated from carrot, pak choi, lettuce, tomato, spinach and cucumber, and were resistant to colistin (CST), AMP, GEN, NA, TET, CIP, cefotaxime (CTX), kanamycin (KAN), levofloxacin (LVX), doxycycline (DOX) and fosfomycin (FOS) [44,45,46]. A study in the Czech Republic found that the majority of *E. coli* isolates (13 of 15) from 108 raw vegetable samples were positive for one or multiple ARGs, including *qac*,* sul1*,* tetA*,* int*,* sul1*,* sul3*,* mer* and *tetB* [10]. One study in Germany reported that 7 out of 245 vegetable samples were positive for extended-spectrum β-lactamase (ESBL)-producing *E. coli*, and all ESBL-producing isolates were positive for *bla*_CTX-M_ genes conferring resistance to third generation cephalosporins (3GC) [11] (Appendix A).

ESBL-producing *K. pneumoniae* recovered from vegetable and fruit samples in Algeria were positive for multiple beta-lactamase genes, including *bla*_CTX-M-15_, *bla*_OXA-1_, and *bla* _SHV-101_, as well as genes that confer resistance to sulfonamides (*sul1, sul2*), tetracyclines (*tetA*), fluoroquinolones (*qnrS1, aac(6′)Ib-cr, qnrB66*), trimethoprim (*dfrA12, dfrA14*), aminoglycosides (*aph(3′)-Ia, aadA2, strB, strA, aac(6′)Ib-cr, aac(3)-IIa*), phenicols (*catA2*) and macrolides lincosamides streptogramins (MLS) (*mph(A)*) [25]. In addition, a study in China found that *K. pneumoniae* isolated from an orange sample was positive for nine ARGs, including *mcr-1, bla*_SHV-110_, *qnrS1* and *fosA6* [43]. *Pseudomonas* spp. harboring two ESBL-genes, *bla*_TEM-116_ and *bla*_SHV-12_, were detected from vegetable samples in a Japanese study [8].

### 3.3. Potential for Adverse Health Outcomes from the Consumption of Fresh Produce Contaminated with ARB/ARGs

We found 25 articles that discussed health risks from consuming fresh produce contaminated with pathogens resistant to one or more antibiotics. However, the articles included in this review did not conduct any risk assessment of potential foodborne diseases due to the consumption of contaminated fresh produce. Instead, these articles broadly mentioned that consuming raw or minimally processed leafy and non-leafy vegetables can be a potential source of foodborne illnesses and invasive bacterial diseases. Moreover, the consumption of contaminated vegetables without any heat treatment or cooking may allow ARB to survive in the food, and reach the human gastrointestinal passage. Multiple studies reported that raw produce could be a vector for transmitting ARGs to the human commensal intestinal flora [22,27,39,40,41]. One article mentioned that the consumption of raw vegetables contaminated with multidrug-resistant pathogens such as *Klebsiella pneumoniae* increased the risk of sharing ARGs (ESBL/AmpC gene) with resident microorganisms in the gut by horizontal gene transfer [25]. Another article mentioned that *Rahnella aquatilis, P. agglomerans, E. cloacae,* and *C. freundii* might contribute to the spread of ARGs to resident bacteria [27]. The public health risks associated with exposure to ARB, especially 3GC-resistant *Enterobacteriaceae* are diverse, ranging from risk of difficult-to-treat diseases, to colonization and asymptomatic carriage, to mere passage through human intestines by environmental species [6,10]. Only three articles discussed the fact that gut colonization by resistant bacteria can pose a risk of complicated infection among infants, the elderly or individuals with weakened immune systems [25,31,37].

Diarrheagenic *E. coli* strains isolated from contaminated cucumber, lettuce and spinach were found to cause diarrhoea and other foodborne gastrointestinal diseases [22]. Consumption of contaminated fruits was also associated with diarrhea diseases, and enterotoxigenic *E. coli* positive for heat-stable enterotoxin-1 gene *astA* was identified as a causative agent [43]. Another article pointed out that the presence of the mobile colistin resistance (mcr 1) gene in *E. coli* isolates from lettuce samples is a serious public health concern, considering that colistin is a last-resort antibiotic used for the treatment of infections caused by multidrug-resistant bacteria. Similarly, *Salmonella* infection (salmonellosis) is one of the consequences of consuming fresh leafy vegetables reported in a study conducted in Malaysia [9]. In this study both *S*. Weltevreden and *S*. Paratyphi were isolated from leafy vegetables. *S*. Weltevreden causes diarrhoea in tropical regions of low-income countries, and *Salmonella* Paratyphi B. also causes enteric fever and gastroenteritis in humans [9]. A few articles (3 out of 40) mentioned that the emergence of drug resistant *Salmonella* infections linked with the consumption of raw vegetables is alarming since multidrug resistance limits the effectiveness of therapeutic treatments [29,31,47].

Antibiotic resistance in *E. coli* is of particular concern because it is the most common Gram-negative bacterial pathogen causing intestinal and extra-intestinal infections in humans [48]. Apart from Gram-negative bacterial pathogens, fresh produce contaminated with Gram-positive bacteria, such as *Listeria* spp. and *Staphylococcus* spp. could be potential sources of foodborne illnesses [47]. Listeriosis caused by *L. monocytogenes* is dangerous for vulnerable individuals; pregnant women and their fetuses, the young, and the elderly, are susceptible to invasive listeriosis, with fatality rates ranging between 20% and 40% [39]. Moreover, uncooked vegetables contaminated with *P. agglomerans*,* P. fluorescens* and *Rahnella aquatilis* could be possible sources of nosocomial infections in vulnerable patients in the hospital [27].

### 3.4. Pathways of Contamination of Fresh Produce with ARB

We found 22 articles that discussed the potential pathways of contamination of fresh produce with ARB. The most common pathways included cross-contamination both during the pre- and post-harvesting periods (Table 3).

Sources of fresh produce contamination with bacterial pathogens during pre-harvesting are diverse, including but not limited to, soils, irrigation water, or animal manure. *E. coli, Salmonella* spp. and *Staphylococcus* spp. have been detected in agricultural soils [43]. *Pseudomonas* species, due to their presence in environmental reservoirs (e.g., soil and water), are frequently found on vegetables. Leafy and non-leafy vegetables such as carrots are at high risk of contamination with soil-borne bacteria, either from the natural microbiota of the soil, or the manure fertilizer used in soil [33]. Untreated animal manure was the most common cause of pre-harvest spread of ARB in fresh produce [9,12,24,41,42,49]. Leafy vegetables such as parsley and water spinach that grow around swamps or riverbanks can be contaminated with wastewater released into these waterbodies by industries, slaughterhouses, or processing plants [9]. Runoff from cattle farms which contains ARB and ARGs, due to the heavy use of antibiotics in animal feed and treatment, may contaminate irrigation water, which can subsequently transfer ARB to fresh produce. Treated or untreated municipal wastewater is used for irrigation in many parts of the world; the absence of wastewater treatment facilities is a major reason for using untreated wastewater in agricultural farms in low-income countries, increasing the potential risk of contamination of produce with ARB [22] (Table 3).

Improper handling of fresh produce during post-harvest processing, including cutting, washing or sanitizing, transporting, packaging or storing, can also create opportunities for microbial cross-contamination [8,27]. The use of contaminated water during post-harvest washing, and the reuse of wash water, were mentioned as reasons for the contamination of fresh produce with bacterial pathogens [42]. Presence of contaminated soil particles that remain as residues on the fresh produce after harvest were mentioned as a potential source of contamination of vegetables with *Arcobacter* spp. [49]. Poor hygiene and sanitation practices of food handlers are often overlooked when it comes to handling vegetables and fruits in retail markets, although these can also be major sources of contamination [43]. One article mentioned that Staphylococcal contamination of fresh produce has been linked to carriage in nasal cavities of infected food handlers, or agricultural workers [50]. *L. monocytogenes* from animal foods can also cross-contaminate fresh produce during processing or display at marketplaces [39].

### 3.5. Recommendations to Reduce Contamination of Fresh Produce with ARB

Antimicrobial resistance surveillance programs primarily focus on food from animal origins, but monitoring antimicrobial resistance reservoirs in food from non-animal origins is equally important [19,51]. In this review, we found that 14 studies recommended action points to reduce the pre- and post-harvest contamination of fresh agricultural produce. These included general precautions to minimize the emergence and spread of ARB, such as controlling the medical and veterinary use of antibiotics. To prevent pre-harvest contamination of produce with ARB, studies recommended proper manure disposal, treating manure before using it as fertilizer, and improving the quality of irrigation water [22,25,42,52]. To prevent post-harvest contamination, studies suggested that: (1) standard sanitation and hygienic practices should be followed by all stakeholders who are involved in food production and supply chain; (2) unsafe or contaminated water (collected from streams or stored in open containers) used to wash and sprinkle over the fruits and vegetables by the vendors should be regulated and monitored to avoid potential cross-contamination; (3) non-chlorine sanitizers for washing, drying, and wrapping or waxing produce after post-harvest should be applied; and (4), irrespective of market type, holding containers and personal hygiene of vendors should be improved [9,23,25,26,47]. Stakeholders should uphold good agricultural and manufacturing practices to ensure food safety for consumers [39]. Additionally, awareness-building programs related to the health hazards of consuming unwashed or improperly washed fruits and vegetables can be implemented [53]. Thoroughly washing with clean water and using food-grade antibacterial chemicals to dip the raw vegetables for a specified duration may eliminate pathogens, and significantly reduce the microbial load [54].

## 4. Discussion

This scoping review revealed the occurrence of ARB and ARGs on fresh produce consumed raw (leafy vegetables, non-leafy vegetables, root, fruits and sprouts) in diverse settings, including high- and low-income countries. While over 20 different bacterial genera were detected in the 40 studies included in this review, some genera were found to be dominant over others. Vegetable samples were frequently contaminated with *E. coli, Klebsiella* spp., *Salmonella* spp., *Enterobacter* spp. and *Pseudomonas* spp. Resistance to various antibiotics and different ARGs was detected in bacterial isolates from produce.

Among all foodborne pathogens in fresh produce, *E. coli* was predominant due to its ubiquity in nature and ease of detection and isolation [10,23]. However, there was a significant variation in the prevalence of contamination between low- and high-income countries. Studies from developing countries such as Bangladesh, India and Pakistan found a high prevalence of *E. coli* in raw vegetable samples sold in retail markets with 60%, 40% and 34% prevalence, respectively [22,26,47]. On the other hand, *E. coli* prevalence was low on vegetables sold in the markets of developed countries, with 3.1% in the USA and 4.1% in Germany [12,55].

The second most reported foodborne pathogen associated with fresh produce contamination and disease outbreaks was *Salmonella* spp. The most frequently reported produce item contaminated with *Salmonella* was lettuce sold in the retail markets. However, the highest prevalence of *Salmonella enteritidis* was reported in long bean and *Salmonella typhimurium* in dropwort in Malaysia, with a prevalence of 67% and 87%, respectively [31]. Possible reasons could include improper handling and poor hygiene practice at retail markets [17,56,57]; however, the higher level of contamination of dropwort may be related to their cultivation in the banks of ponds where liquid waste from various sources are disposed of [58]. Furthermore, the use of animal faeces as fertilizer has been suggested as another pathway for the contamination of vegetables with antimicrobial-resistant *Salmonella* serotypes from animal origin [59]. Both *E. coli* and *Salmonella* isolates from fresh produce samples were often resistant to multiple antibiotics of clinical importance, including ampicillin, erythromycin, co-amoxiclav, and cephalothin [60].

Contamination of fresh produce by opportunistic bacterial pathogens is a food safety concern, particularly for immune-compromised individuals. Studies in this review found *Enterobacter* spp. including *E. cloacae, E. gergoviae* and *E. faecalis* from fresh produce were resistant to multiple antibiotics including tetracycline, co-amoxiclav and cefotaxime [55]. Although the prevalence of *Klebsiella* species in fresh vegetables was variable in different studies, multidrug-resistant strains from produce samples were associated with clinical infections in humans including bacteremia, septicemia and urinary tract infections [25,43,52]. Therefore, the surveillance of foodborne pathogens in fresh produce samples should include this microorganism in the panel of other bacterial pathogens.

Our review found that contamination occurred during both pre- and post-harvest processes. Produce can be contaminated at any point in time from the farm to the retail markets: pre-harvest via untreated manure, untreated irrigation water, contaminated soil; post-harvest during handling at the farm, transportation and storage [61], and even during the time left out on display in retail markets. The increasing amount of antibiotics used in animal feeds and as veterinary drugs affects entire agro-eco systems, and threatens raw vegetable production systems, the soil ecosystem and the quality of groundwater [62,63]. Unsafe or contaminated water used for washing, spraying and dipping also contaminates fruits and vegetables during post-harvest processing [64], which needs to be regulated by monitoring and implementing good agricultural practices. Re-use of water used for washing and contaminated soil particles can also cause cross-contamination [65]. Poor personal hygiene by vendors and agricultural workers [66], poor sanitation facilities [67], and unhygienic conditions at marketplaces [54] are also associated with the contamination of fresh vegetables and fruits with ARB. Some types of fruits and vegetables in the retail markets may be more likely to be contaminated with ARB, due to being constantly handled by shoppers to check freshness or ripeness, although no evidence exists to prove this mechanism [55]. Unhygienic or neglected practices in the preparation of fresh-cut produce, packaged RTE vegetables or mixed salads can thus increase the risk of foodborne diseases, instead of offering healthy lifestyle choices [61].

## 5. Limitations

Our review identified knowledge gaps which require further research to understand the magnitude of the public health risks from transmission of ARB/ARGs through fresh produce. From the articles included in our review, it was not possible to determine the extent of the human health risk associated with the consumption of fresh vegetables and fruits containing ARB, due to the absence of dose–response data, and epidemiologic investigations. Studies have attempted to quantify human exposure to ARB via irrigated produce [68]; however, quantitative assessments of health risks cannot be conducted without an understanding of the dose–response relationships for ARB, which differ from traditional pathogens. Additionally, no study in our review reported epidemiological analyses on health endpoints, such as gut colonization with ARB or clinically confirmed infections among the consumers of contaminated fresh produce. This is consistent with the findings of a recent review conducted for risk assessment purposes [3]. Similarly, most of the studies explored the prevalence of antimicrobial-resistant pathogens, resistant strains or serotypes, or resistance genes, but potential infections related to those specific pathogens were not reported in the articles.

Other limitations are that some studies focused on specific bacterial pathogens on samples, and therefore no information is available on the presence of other foodborne pathogens. Studies frequently reported the detection of ARBs, but not their abundance in produce. Only a subset of studies detected ARGs and similarly focused on detecting specific genes. There was a lack of information on whether the studies followed standard guidelines for antibiotic susceptibility testing. Resistance profiles of the same bacteria varied between studies due to different location/countries, types of samples (vegetable, fruits, and leafy greens), choice of antibiotics investigated, detection/identification methods (disk diffusion, dilution, e-test), and breakpoints and standards to determine resistance (e.g., European EUCAST organization vs. U.S. Clinical and Laboratory Standards Institute). Due to this heterogeneity across studies, a statistical meta-analysis of the data was not conducted. Finally, since we only included articles written in English, publication bias may influence our findings.

## 6. Conclusions

The presence of ARB and ARGs in fresh produce and RTE salads consumed raw poses potential public health risks of unknown magnitude. Preventing ARB/ARG exposure through fresh produce may be challenging considering the cross-cutting issues related to food security and food safety, accentuating the need for a holistic approach encompassing pre-harvest and post-harvest processing and the distribution of produce. In addition to systematic surveillance and monitoring, communication on necessary behavioural changes and awareness-raising programs are needed for all stakeholders, including farmers, handlers involved in transport and storage of produce, retailers, and consumers. Future studies should be directed towards quantifying human exposure to ARB/ARGs via fresh produce, and assessing the associated health risks among consumers.

## Figures and Tables

**Figure 1 ijerph-19-00360-f001:**
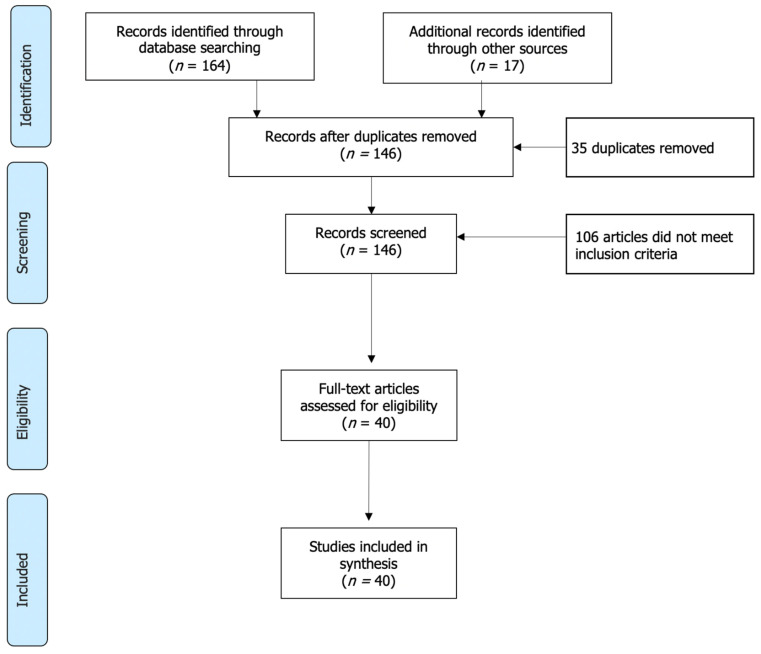
PRISMA flowchart illustrating the study selection process.

**Table 1 ijerph-19-00360-t001:** Literature search strategy for scoping review.

Title	Contamination of fresh produce with antibiotic-resistant bacteria and associated risks to human health: a scoping review		
Research question	What is the presence and abundance of ARB, ARGs and antimicrobial residues on fresh produce sold in the retail markets, and how do they affect human health?		
Search Strategy	Inclusion Criteria	Studies that detect and/or quantify ARB, ARGs and antimicrobial residues on fresh produce (vegetables/leafy green/fruits) sold in retail markets (e.g., vendors, supermarkets, farmer markets)	
		Types and abundance of antimicrobial residues present on fresh produce	
		Pathways for ARB, ARGs and antimicrobial residues entering fresh produce	
		Health risks associated with consumption of fresh vegetables, leafy greens or fruits contaminated with ARB, ARGs and antimicrobial residues	
		Full-text peer-reviewed journal articles and grey literature	
		Species: Human	
		Language: English	
	Exclusion Criteria	Articles that did not include fresh agricultural product consumption and its relationship with AMR	
		Articles that analysed AMR with relation to mixed or ready-to-eat salads with various dressings	
		AMR-related human health risks from exposures other than fresh produce	
		Animal-based foods (e.g., chicken, beef, pork, eggs, milk)	
		Animal agriculture (e.g., poultry, meat, dairy, fishery)	
		All types of review articles	
	Time Frame	1 January 2001–18 October 2020	
Data Sources	Peer-reviewed articles	Ovid Medline, Web of Science, Hinari	
	Grey literature	Google, Google Scholar, Proquest	
Key search terms	Antimicrobial Resistance-related terms (combined by ‘OR’) (a)	Agriculture and fresh agricultural products-related terms (combined by ‘OR’) (b)	Place of items/sample collected (c)
	Antimicrobial resistance, antimicrobial residues, antibiotic-resistant bacteria, antibiotic resistance genes, antimicrobial-resistant organisms, antibiotic-resistant pathogens, health risks	Agriculture, farming, fresh agricultural produce, fresh agriculture products, fresh vegetables, raw vegetables, salad vegetables, leafy greens, fruits	Retail markets

**Table 2 ijerph-19-00360-t002:** Framework for data analysis.

Theme	Sub-Theme
Presence and abundance of ARB, ARGs and antimicrobial residues on fresh produce (raw consumed vegetables, fruits) sold in retail markets	Prevalence of antimicrobial-resistant pathogens on the fresh produce
	Strains/serotypes of antimicrobial-resistant bacteria on fresh produce
	Antimicrobial resistance genes on fresh produce
Public health risks from consuming raw agricultural products or fresh produce in relation to AMR	
Pathways of contamination of fresh produce with ARB/ARGs/antimicrobial residues	
Actions recommended to reduce the contamination of fresh produce with ARB, ARGs and antimicrobial residues	

**Table 3 ijerph-19-00360-t003:** Potential pathways of contamination of fresh produce with antimicrobial-resistant bacterial pathogens.

Pathogens	Pre-Harvesting (Number of Articles)	Post-Harvesting (Number of Articles)
*Salmonella* spp. *E. coli* *Arcobacter* spp.	Use of untreated animal manure from livestock (6). Use of untreated wastewater in irrigation (5). Contaminated irrigation water by runoff from cattle farms (4). Contaminated soil with pathogens or animal faecal material (3). Compromised environment for growing leafy vegetables (3).	Improper handling of fresh produce during post-harvest process (2). Post-harvest washing with contaminated water (2). Poor hygiene and sanitation practice of food handlers (2). The presence of soil particles in vegetables contaminated with animal faecal material (1).
Quinolone-resistant *Salmonella* spp.	Human or poultry farm wastewater or sewage mixing with irrigation water used in vegetable production (1).	
*Staphlococcus aureus*		Infected food handlers or workers (1).
*Listeria monocytogenes*		Cross-contamination from animal foods (1).

## Data Availability

Data is available upon request.

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
