# Peer review of "Contamination of Fresh Produce with Antibiotic-Resistant Bacteria and Associated Risks to Human Health: A Scoping Review"

_ijerph, 2021, doi:10.3390/ijerph19010360_

Round 1
Reviewer 1 Report
The aim of this scoping review was to provide useful information about antimicrobial-resistant bacteria (ARB) and their resistance genes distribution in fresh agricultural products consumed raw. By making a comparison between databases for peer-reviewed literature (Ovid Medline, Web of Science and Hinari websites) and grey literature, authors found that E. coli, Klebsiella spp. and Salmonella spp. were the most commonly ARB observed on lettuce. Moreover, paper suggested recommendations to reduce contamination with antibiotic-resistant microrganisms during pre- and post- harvest processes of food products.
The topic is of this study is interesting and timely. The manuscript is well written and structured:it should be of great interest to the readers. In general, I find the manuscript very clearly written. English is fairly good and the text flows well. Neverless, there are many typos throughout the manuscript. The paper is acceptable for publication with minor revisions.
Some specific comments
Introduction
page 2
- line 78: please explain AMR acronym when first used
Results
page 7
- line 156: please add a copy of the sentence “The most commonly….Salmonella spp. (20%)” from lines 32-33 to the beginning of the paragraph “Prevalence of antimicrobial-resistant bacteria on fresh produce”.
- line 163: please add DEPs acronym after “diarrheagenic E.coli”
- line169: please add DEPs acronym after “Pakistan tested 50”
- line 171: please add “not DEPs” after E.coli
- line 171: please explain AMX acronym
- line 172: please explain CST acronym
- line 173: please explain CAZ acronym
- lines 178-180: results reported do not refer to reference number 27
- line 182: please explain ATM acronym and change to lowercase the first letter of “were”
- line 184: please explain FOX acronym
- line 186: please change “eight” to “nine”
- line 193: please change “Tuscon” to “Tucson”
- line 197: please delete “amoxicillin”
- line 198:please change reference number from 36 to 46. In Table S4, resistance to AMP, AMX and TMP of S.enteritidis was not indicated: please correct it.
- lines 199-201: please mark all different salmonella species in italics
- lines 202-203: please mark all different salmonella serotypes in italics
- line 206: please explain STR acronym
page 8
- line 217: please change “six” to “five”
- line 221:please change reference number from 36 to 69
- lines 223-224: please mark E.coli in italics
- line 224: please add AMG acronym after “aminoglycosides”
- line 229:please change reference number from 39 to 46
- line 231: In Table S4, resistance to ERY, OXA, CEF, VCM, STR and CIP of L. monocytogenes was not indicated: please correct it.
- lines 234-242: these sentences should be move to the beginning of the paragraph (line 156)
- line 235: please mark E.coli in italics
- line 236: please change “E.cloaca” to “E. cloacae”
- line 237: please mark Staphylococcus spp., Yersinia enterocolitica in italics. In Table S4 Yersinia enterocolitica was not cited: it is a mistake? Please correct it
- lines 239-241: please delete “S.multivorum, P.agglomerans and E.faecium”: they are redundant
- line 240: please change “baumanni” to “baumannii”
- lines 247-251: please mark E.coli in italics
- line 249: please delete “where tet A and tet B”: it is redundant
- line 254: please change “CT” to “CST”
- line 257: please delete “of majority”: it is redundant
- line 259: please change “6” to “7”
page 9
-line 305: please mark E.coli in italics
-lines 309-310: please mark salmonella species in italics
- line 312: please change to lowercase the first letter of “paratyphi”
page 10
-lines 364-365: please delete (Error! R eference source not found.).
page 12
-line 418: please add a reference about Salmonella study in Malaysia cited
-line 431: please change “Cefotaxime” to “cefotaxime”
Table S4
- please change the first letter of bacteria species to lowercase (see references 9, 46,33,32)
- page 25: please delete “P.paralactis” (reference 8):it is a repetition
- page 25: change “Yingjiao Li et al 2019” to “Yingjiao Li et al 2020”
- page 26:please change reference 69 to 36
- page 26:please delete “CTX” in reference 45 and S.multivorum in reference 13.They are repetitions
- page 27: please change “baumanni” to “baumannii” (reference 13) and “Hamilton-miller” to “Hamilton-Miller”(reference 70)
- page 28: please change “Hikmate A” to “Abriovel H” (reference 42) and change “Imipenm” to “Imipenem” (reference 51)
- page 29:please delete “Nigad” in “MeherNigad Nipa” (reference 53)
References
- please complete author's first name with AM in reference number 6
- please add Janalikova M as first author in reference number 10
- please add page number (52) in reference 14
- please delete “2016” in bold in reference 16
- please add issue number (1) in reference 19
- please move year of publication (2009) after the journal title and change “108” to “1108” in reference 21
- please write the first author's name in lowercase letters in references 31 and 34
- please change the first author's name from “M” to “M.S.” in reference 33
- please delete “2012” in bold in reference 36
- please add issue number (1) and page numbers (13-18) in reference 51
- please write the journal title in lowercase letters and add issue number (4) in references 57
- please delete the second “50” in page numbers in reference 64:it is redundant
- please add page number (429) in reference 67
- please delete references 69 and 70: they are a repetition of 36 and 27 references, respectively.
Author Response
Thank you so much for your appreciation, observation, and detailed comments. We have addressed all your comments and suggestion in the introduction (page number 2), results (page number 7 to 10), discussion (page number 12), conclusion (page number 13), Table S4 and references.
In the result section, (page7, line 202-203) we did not mark all the salmonella serotype italic since CDC does not recommend serotype names as italics "The name of the genus, species and subspecies should be italics and the name of the serotype should be non-italics starting with a capital letter such as Salmonella enterica subsp. enterica serotype Typhi" (Ref: Brenner FW, Villar RG, Angulo FJ, Tauxe R, Swaminathan B. Salmonella nomenclature. J Clin Microbiol. 2000 Jul;38(7):2465-7. doi: 10.1128/JCM.38.7.2465-2467.2000. PMID: 10878026; PMCID: PMC86943.)
In the page 8 line 224, we did not add acronym of aminoglycosides since aminoglycoside is an antibiotic class and usually acronyms are not used for classes of antibiotics.
In the page 8, line 217, we carefully checked our findings and found that six of the included studies reported the isolates of Pseudomonas spp. resistant to AMP, therefore we did not change the number.
In the line number 223-234, we mentioned E. cloacae, not E. coli that have changed to the italic form. We moved the sentences (line 234-242) where appropriate.
In the line 237, Yersinia enterocolitica was mistakenly mentioned which is now deleted.
Thank you so much for your suggestions.
Reviewer 2 Report
This review investigated literature reporting contamination of fresh produce, primarily vegetables, with antibiotic resistant microorganisms. The selection criteria for inclusion of searched literature appears to be too strict as there is more published information that should be included in the review. The review claimed that there were few studies reporting antibiotic-resistant microorganisms and their associated genes in fresh produce, but I believe that there is considerably more information that was not included in this review.
What were the grey literature sources that were included? The potential for adverse health outcomes section needs to be expanded. For example, listeriosis is not mentioned. The review is otherwise well written. The pathways to contamination section nicely summarises this information.
Specific comments
The abstract is too long. It needs to be a maximum of 200 words.
Line 53, 66-67 – This is incorrect. There are many studies reported. A quick search of CAB abstracts (Web of Science, which was indicated as being searched) using the key words of “antibiotic resistance + vegetable” yielded 1200+ results; antibiotic resistance gene + vegetable = 475; antibiotic resistance + vegetable + retail = 64; antibiotic resistance + vegetable + ready-to-eat = 53; antibiotic resistance + vegetable + fresh = 145; antibiotic resistance + vegetable + market = 112; antibiotic resistance + vegetable + raw = 144. Further articles would most likely be found using other related key words. In your exclusion criteria, you indicate that articles that did not mention agricultural product consumption were excluded. This reduces the available studies considerably. Please be careful using broad statements such as those in lines 53 and 66-67. Those statements need to be qualified to indicate why there are not many studies. As these sentences are written, it suggests that not many studies were conducted at all. This is incorrect. There were many studies; however, they may not all meet your selection criteria.
Line 134 After duplication removal (not remove)
Line 134 Literature search from various (not form)
Line 135 Only 146 unique articles were found prior to exclusion for not meeting inclusion criteria? This seems remarkably few articles.
Line 147 antibiotic resistant microorganisms (not organism). Throughout manuscript, please change “organism” for “microorganism”.
Lines 173, 224, 235, 237, 247, 249, 251, 305 Please make sure that scientific names are in italics. Please check the entire manuscript.
Line 181 “Leafy vegetable lettuce was mostly contaminated”. This is a vague sentence. What is “mostly”? Please make it more specific.
Line 211 Change included to including
Line 236 E. cloacae, not E. cloaca.
Line 249 Please rewrite this sentence or end it after “and radish samples”.
Line 310 were isolated from leafy vegetables
Line 310-311 S. Weltevreden was the most commonly found serovar of Salmonella in that study, not necessarily the most common serovar for all studies. Please clarify.
Line 340 vegetables such as parsley and water spinach
Line 363 Listeria monocytogenes is spelt incorrectly
Line 364 missing reference
Line 399 bacterial genera
Table 3 How does column 1 correlate with columns 2 and 3?
Author Response
Thank you so much for your observation and detailed comments.
We edited the abstract and keep it within 200 words. We also edited the line 53, 66-67 in the introduction section.
In the result section, line 147 and throughout the manuscript, we have changed antibiotic-resistant microorganism to antibiotic-resistant bacteria where appropriate.
We have checked the entire manuscript and made the scientific name italics.
We have addressed all your comments and suggestion in the result section (line 134, 173,181, 211, 224, 235-236, 237, 247, 249, 251, 305, 310, 311, 340, 363 and 364).
In table 3 (page 11), the second and third columns represent the pathways of contamination of fresh produce in relation to the antimicrobial pathogens in first column. Grey literature sources are already included in the table 1. In the health outcome section, we only discussed those health risks mentioned by the authors of the included articles. We have added the adverse health outcome of Listeriosis.
We are grateful for your extensive review and great suggestions.
Reviewer 3 Report
The manuscript discuss one of the most important issue of recent times. Article selection and data compilations are well executed . There are minor grammatical and/sentence re structing issues that are to be taken care of. The minor correction request is not restricted the the line numbers mentioned below:
Line159-160, 183, 193, 206-208, 211319-321,335-336,338-339
Line 184: Include Ready-to-eat for RTE which is mentioned later in the article .
If possible please reconsider to restructure some of the sentences as they sound bit repetitive.
Author Response
Thank you so much for your comments and suggestion. We have checked all the mentioned lines (Line 159-160, 183, 193, 206-208, 211,319-321, 335-336, 338-339) and the entire manuscript, and edited where appropriate.
Thanks again for your suggestions.